# Influence of a Biocompatible Hydrophilic Needle Surface Coating on a Puncture Biopsy Process for Biomedical Applications

**Fan Gao [1], Qinghua Song [1,2,*], Zhanqiang Liu [1,2], Yonghang Jiang [1] and Xiuqing Hao [3]**

[1] Key Laboratory of High Efficiency and Clean Mechanical Manufacture, Ministry of Education, School of Mechanical Engineering, Shandong University, Jinan 250061, China; gfxc113@163.com (F.G.); melius@sdu.edu.cn (Z.L.); jiangyh1994@foxmail.com (Y.J.)

[2] National Demonstration Center for Experimental Mechanical Engineering Education,Shandong University, Jinan 250061, China

[3] College of Mechanical and Electrical Engineering, Nanjing University of Aeronautics and Astronautics, Nanjing 210016, China; xqhao@nuaa.edu.cn

* Correspondence: ssinghua@sdu.edu.cn; Tel.: +86-531-88392045

**Abstract:** A puncture biopsy is a widely used, minimally invasive surgery process. During the needle insertion process, the needle body is always in direct contact with a biological soft tissue. Tissue adhesion and different degrees of tissue damage occur frequently. Optimization of the needle surface, and especially the lubrication of the needle surface, can deal with these problems efficiently. Therefore, in this paper, a biocompatible hydrophilic coating was applied onto the surface of a needle to improve the surface quality of the needle surface. Further, a simplified finite element model of insertion was established, and extracorporeal insertion experiments were used to verify the accuracy of the model. Then, by analyzing a simulation model of a coated needle and a conventional needle, the influence of the application of the coated needle on the insertion process was obtained. It can be seen from the results that the coating application relieved the force on the needle and the soft tissue during the insertion process and could significantly reduce friction during the insertion process. At the same time, the deformation of biological soft tissue was reduced, and the adhesion situation between the needle and tissue improved, which optimized the puncture needle.

**Keywords:** coating; puncture needle; tissue adhesion; surface optimization; simulation

## 1. Introduction

In modern medical technology, minimally invasive surgery and local treatment surgery play an important role. As a minimally invasive interventional surgery, puncture biopsy has the advantages of being less traumatic and having strong targeting, an aspect common in modern surgery, such as in tissue biopsies and targeted drug therapy [1–4]. Puncture biopsy sampling is the main method to obtain tumor tissue and make a pathological diagnosis [5]. A puncture needle is the main tool of a puncture biopsy, and has been widely used in clinical practice. In order to make an accurate medical diagnosis of a pathological tissue, it is crucial to obtain large and exact biopsy samples. In practice, to protect healthy tissue, doctors have detailed control over the position and speed of the needle [6]. Therefore, many former researchers have investigated simulation methods with respect to puncture biopsies, which can provide reference for the actual puncture operation. Zhao et al. [7] used Ansys software to analyze soft tissue deformation during the insertion process. They simulated acupuncture by applying force to a point in biological soft tissue. Aneissha et al. [8] conducted a simulation analysis on the force of the puncture needle. Silviu et al. [9] fixed organs using elastic virtual

elements that allowed for simulating a link with other organs. However, they studied the needle body and the biological soft tissue independently, without considering the actual interaction between the needle body and the biological soft tissue. Later, Peng et al. [10] and Gu et al. [11] established a two-dimensional puncture model to simulate the process of needle insertion using ABAQUS. However, during this process, the needle body did not react with force. Yen et al. [12] and Lin et al. [13] used cohesive elements arranged along a predefined crack path that the needle would pass through. The tip of the needle always stayed on the path because the nodes on the path were constrained properly. Mohamed et al. [14] used gel instead of soft biological tissue and developed a three-dimensional finite element (FE) model to simulate needle–tissue interaction during needle insertion. They used a linear elastic model to predict the needle interaction. Because complex components in biological soft tissue lead to the complexity and diversity of its mechanical properties, the linear elastic model is a simplification that cannot represent the real situation of biological soft tissue.

Furthermore, the surface morphology of the puncture needle also has an important impact on the quality of the puncture. Andrea et al. [15] found that using a helical-tip needle in surgery can lead to diagnostic samples with less fragmentation, greater weight, and lower length, comparable to a traditional needle. It can provide a higher amount of adequate tumor tissue samples even with shorter samples. Frasson et al. [16] printed a microtexture on the outer surface of a neurosurgical probe. They found that it could significantly affect the insertion and extraction forces generated at the tissue–needle interface. Andreas et al. [17] developed a novel type of neurosurgical probe with surface texturing and various microstructure geometries on the needle body. Then, they investigated tribological properties during an insertion of the probe into brain tissue. Tassanai et al. [18] discovered that adding a microtexture to the surface of a deep brain simulation (DBS)-like probe can minimize the migration of an electrode in the extracorporeal porcine brain, which could be a method for reducing DBS lead migration without additional tissue damage. Chen et al. [19] prepared polymer-coated microneedles for drug delivery and studied the mechanical properties, transdermal drug delivery effects, drug loading, and drug delivery efficiency of the microneedles. Asad et al. [20] coated microneedles uniformly with a thin porous polymer (poly(lactic-co-glycolic acid), PLGA) and investigated the effect of coating applications. They found that it could deliver drugs at high rates and reduce the deformation of microneedles during piercing of the skin. Lo et al. [21] developed a manufacturing process for coating the nerve probe with an ultrafast degradable polymer. The coating provided sufficient rigidity to the inserted probe, and at the same time, the coating rapidly degraded within a few hours without harm to the human body. This made it possible to record neural signals over a long period of time.

During the puncture biopsy process, the quality of the needle surface is closely related to the wound shape formed during surgery and the adhesion of biological soft tissue to the needle body. In existing studies, some researchers have proposed that the application of surface microtextures and coatings on a microneedle surface could improve drug transport efficiency and enhance the stiffness of the needle body. However, when the size of the microtexture extends beyond a certain range, it will cause serious damage to the tissue. In this paper, a kind of biocompatible hydrophilic superlubrication coating was applied to the surface of a puncture needle body. A friction coefficient measurement experiment was designed to measure the friction coefficient of the coated needle. In this study, a simple method of establishing a puncture model using ABAQUS is proposed, and the accuracy of the model was verified with extracorporeal insertion experiments. Then, the model was used to analyze the influence of applying a coated needle during a puncture biopsy compared to using a conventional needle.

## 2. Materials and Methods

### 2.1. Selection of Experimental Needle and Coating

In the experiments and simulation, a widely used conventional puncture needle was selected that had a 1.6-mm outside diameter, a 150-mm length, and a 17° tip angle. Figure 1 shows the appearance and tip structure of the puncture needle.

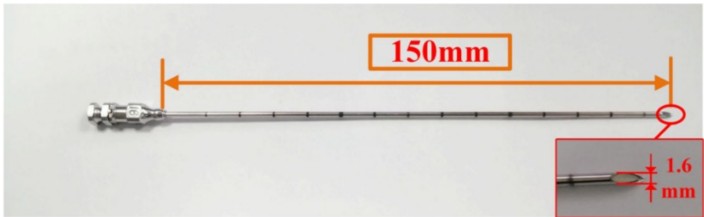

**Figure 1.** The puncture needle.

The coating solution consisted of polyvinyl pyrrolidone (PVP) as the main solute and ketones as the solvent. Neither of these materials would react chemically with the needle material. When in contact with water, the biocompatible hydrophilic superlubrication coating would turn into a hydrogel-like material. The coating passed the tests of cytotoxicity, sensitization, and irritability. Moreover, during use, there was slight shedding, but it passed a national evaporation residue test. The coating method included spray and dip coating, and the thickness was approximately 2–8 μm. When the coating was applied to the surface of the needle body, because the coating material was liquid and its thickness was on a micron scale, it was difficult to detect with the naked eye. By means of a metallographic microscope, the surface microstructure morphologies of the coated needle and the conventional needle under a microscope are shown in Figure 2a,b, respectively. It can be observed that on the surface of the conventional needle body, there were some wire-drawing grains left caused by metal machining processing. Meanwhile, on the surface of the coated needle body, there were also bubbles formed after contact with liquid, and the reflection was obvious. This indicated that the coating on the surface of the needle body would not have a great impact on the shape and structure of the needle body.

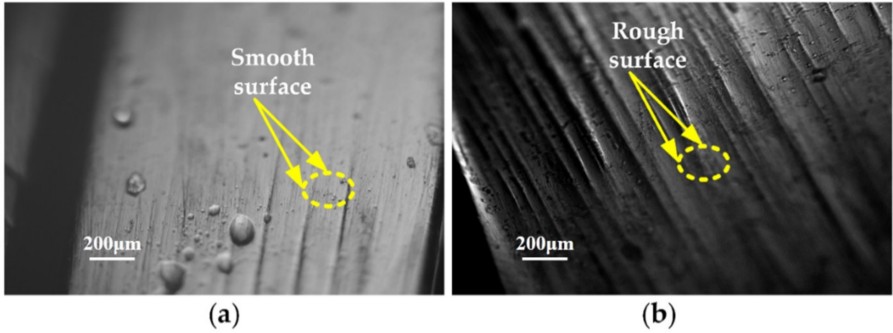

**Figure 2.** Microscopic view of the needle surface: (**a**) surface morphologies of the coated needle; (**b**) surface morphologies of the uncoated needle.

### 2.2. Experiment Samples

In an abdominal puncture biopsy, in the path of the puncture needle there is no thick epithelial tissue or connective tissue, such as a tendon. Therefore, when selecting the experimental samples, the outer membrane of a muscle and large blood vessels or connective tissues should be avoided as far as possible. When a puncture biopsy surgery is performed, the activities of biological soft tissue have much to do with its mechanical properties. Hence, it is very important to ensure the biological activity

of experimental samples during extracorporeal insertion experiments. To make sure the biological soft tissue used in the experiment was consistent with the model in the simulation, fresh porcine loin tissue was selected as the sample for the insertion test. This was the same material used in the uniaxial tension test. Generally, muscles are mainly composed of many small muscle fibers, as can be seen in Figure 3a, and the connective tissue and blood vessels between them can be seen in Figure 3b [22]. Tiny blood vessels are randomly distributed among muscle fibers and connective tissue, which are too small to be pointed out in Figure 3.

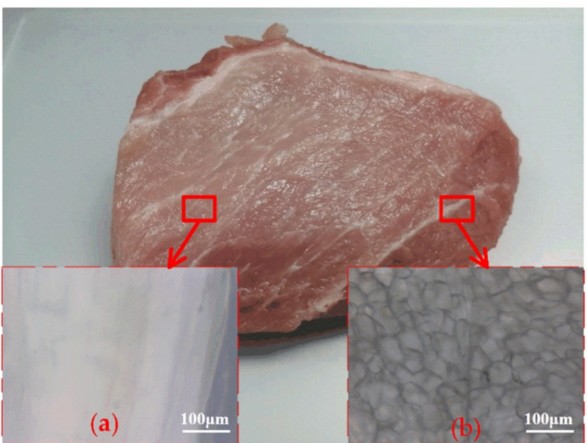

**Figure 3.** Major components of biological soft tissue: (**a**) muscle fibers; (**b**) connective tissue.

### 2.3. Simulation Method for Tissue

In a puncture biopsy, the path of the needle will pass through many complex biological soft tissues. Biological soft tissue includes skin, subcutaneous tissue, muscle, nerves, blood vessels, and other human tissue. Epithelia, connective tissue, muscle, and nerve tissue are typical soft tissues in the human body [23]. Collagen fibers, reticular fibers, and elastic fibers in the extracellular matrix are the main reason for the elastic properties of biological soft tissues [24]. Researchers have diverse interpretations of the nonlinear phenomena and many complex behaviors of biological soft tissues [25]. Biological soft tissues can support large deformations, and their mechanical behavior depends primarily on time and previously sustained maximum deformation [26,27]. At present, the constitutive equations of most biological soft tissues are described by simulating biological soft tissues as hyperelastic materials [28].

The deformation behavior of a biological soft tissue can be described using a strain energy density function [28,29]. Constitutive equations describing biological soft tissue are mostly based on continuum mechanics, i.e., the strain energy density function depends on the three strain invariants $I_1$, $I_2$, and $I_3$ and $J$ in the right Cauchy–Green deformation tensors. Some unique characteristics of biological soft tissues, such as heterogeneity, nonlinearity, and compressibility, are revealed in different minimally invasive procedures. Generally, the overall volume of organs will remain basically unchanged in a minimally invasive surgery, which means the volume of biological soft tissue changes slightly during the process of puncture biopsy. Therefore, it is assumed that all biological soft tissues are approximately incompressible. $I_3$ is considered constant and equal to 1, which means that $I_3$ does not contribute to the strain energy. The most classical strain energy density formula can be expressed as a polynomial composed of invariants [28]:

$$W = W(\bar{I}_1, \bar{I}_2, J) = \sum_{i+j=1}^{N} C_{ij}(\bar{I}_1 - 3)^i(\bar{I}_2 - 3)^j + \sum_{i=1}^{N} \frac{1}{D_i}(J - 1)^{2i} \tag{1}$$

where $W$ is the strain energy density formula, $I_1$ and $I_2$ are the first two principle invariants of the right Cauchy–Green deformation tensors, $J$ is the deformation gradient tensor determinant for incompressible biological soft tissue materials $J = 1$, and $C_{ij}$ and $D_i$ are material constants. In this paper,

through fitting with a uniaxial tensile experiment, the Mooney–Rivilin model was selected to describe the deformation behavior of biological soft tissue in the simulation [29]. The material constants of the biological soft tissue model were obtained by considering hyperelasticity, plasticity, and the fracture mechanical properties of the biological soft tissue and by matching the data of previous biological soft tissue uniaxial tensile experiments. The Mooney–Rivilin strain energy density formula is given by

$$W = C_{10}(I_1 - 3) + C_{01}(I_2 - 3) \tag{2}$$

where $C_{10}$ and $C_{01}$ are the mechanical constants of the experimental material according to the experimental data given by the uniaxial tensile test, and $I_1$ and $I_2$ can be obtained from

$$I_1 = \lambda_1{}^2 + \lambda_2{}^2 + \lambda_3{}^2, I_2 = (\lambda_1\lambda_2)^2 + (\lambda_2\lambda_3)^2 + (\lambda_3\lambda_1)^2 \tag{3}$$

where $\lambda_i$ ($i$ = 1–3) represents the elongation ratios. The stress–strain relationship characterizes the main properties of the material. It can be expressed by the partial derivative of the strain energy density function with respect to the principal elongation ratio. For uniaxial loading, the stress–strain behavior relationship of the Mooney–Rivilin model is given by

$$\frac{t_1}{2\left(\lambda_1 - \frac{1}{\lambda_1{}^2}\right)} = C_{10} + \frac{1}{\lambda_1}C_{01} \tag{4}$$

According to the stress value $t_1$ under different tensile ratios $\lambda_1$ measured in the experiment, $\frac{1}{\lambda_1}$ is taken as the abscissa, and $\dfrac{t_1}{2\left(\lambda_1 - \frac{1}{\lambda_1{}^2}\right)}$ is taken as the ordinate. The test points obtained in the experiment are plotted in this coordinate system, and the discrete experimental datapoints are fitted onto a straight line. $C_{10}$ is the intercept of this line, and $C_{01}$ is the slope of this line. The parameters of the biological soft tissue were gained from experiments, and other material constants are shown in Table 1.

**Table 1.** The parameters of the biological soft tissue used in the simulation.

| Density (kg/m$^3$) | $C_{10}$ | $C_{01}$ |
|---|---|---|
| 1120 | 4.75 | −4.91 |

### 2.4. Measurement of the Coefficient of Friction (COF) between the Coated Needle and Biological Soft Tissue

In this study, a new coating needle was used. Although the application of a coating does not have much effect on the shape of the needle body, it still affects the surface properties of the needle. In particular, the friction coefficients of contact between the needle and the tissue will be changed. In this section, to get the friction coefficient of contact between the coating needle and the tissue in the insertion process, a friction coefficient measurement experiment was carried out. The friction coefficient between the coated needle and the biological soft tissue was measured using the experimental apparatus shown in Figure 4.

A flat piece of biological soft tissue was inserted under the coated needle, and the pressure between the coated needle and tissue was measured by a six-axis force and a torque sensor (ME-Meßsysteme GmbH, Hennigsdorf, Germany). The data collection frequency was 800 Hz, with the pressure set to $N$, and the force in the $Z$ and $Y$ directions was measured by a six-axis force and a torque sensor. Using the experimental data, the pressure $N$ of the needle body on a biological soft tissue surface can be expressed as

$$N = \sqrt{F_z{}^2 + F_y{}^2} \tag{5}$$

The magnitude of the $x$ axis force measured is $F_x$. After the analysis, it was found that the $F_x$ measured here is friction force. Therefore, the friction coefficient is expressed as follows:

$$\mu = \frac{F_x}{N} \tag{6}$$

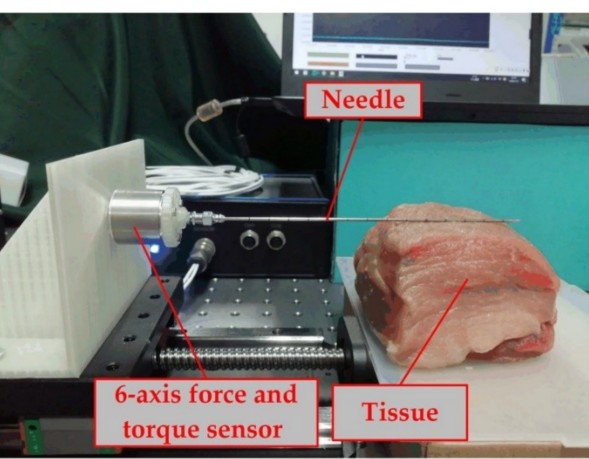

**Figure 4.** Friction coefficient measuring device.

In Figure 5, the different data obtained through experiments are fitted onto a straight line, and the slope of this line describes the measured friction coefficient. The force condition after the insertion may be more complicated than that in this experiment because of the complex mechanical conditions of a biological soft tissue, including elastic plastic strain and fracture [30]. Therefore, in this study, the tiny error marked in Figure 5 was ignored, and the friction coefficient was set as 0.03, which was within the data range given by the coating provider (Chengdu DAXAN Innovative Medical Tech. Co., Ltd., Chengdu, China). Because the lubrication function of the coating was very stable, the measured data were well linear.

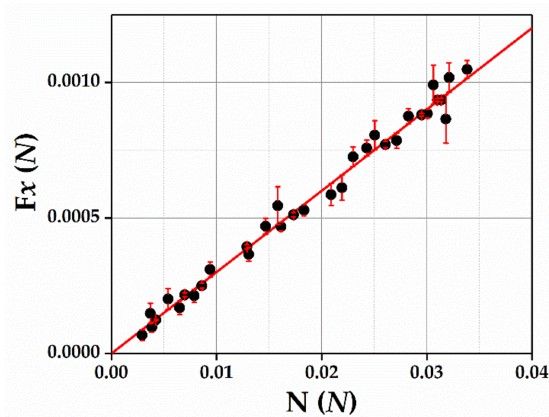

**Figure 5.** Experimental data for the measurement of the friction coefficient.

## 3. Simulation Model Verification

### 3.1. FEM Model Development

For ease of illustration and a comparison of the results, a two-dimensional dynamic model developed in ABAQUS 6.14 (Dassault Systèmes, Vélizy-Villacoublay, France) and an explicit solver were used to simulate the needle insertion process. The parameters of the tissue model came from Table 1, and the shape and size matched the experimental samples. Here, $x$, $y$, and $z$ rotation and $x$, $y$,

and $z$ translation were constrained along the bottom edge of the biological soft tissue model. Since the deformation of biological soft tissue is large deformation in the needle insertion process, Arbitrary Lagrangian Eulerian (ALE) adaptive mesh was adopted for the model. Tissue separation was defined by setting the fracture strain and displacement at failure in ABAQUS. It can be seen in Figure 6 that a local-mesh refined method was used to the tissue model. The elements on the center of the biological soft tissue were refined to obtain a more accurate numerical solution in the high-stress and strain gradient zone. Nevertheless, the elements on the border of the biological soft tissue were coarsened to reduce the time of calculation as much as possible.

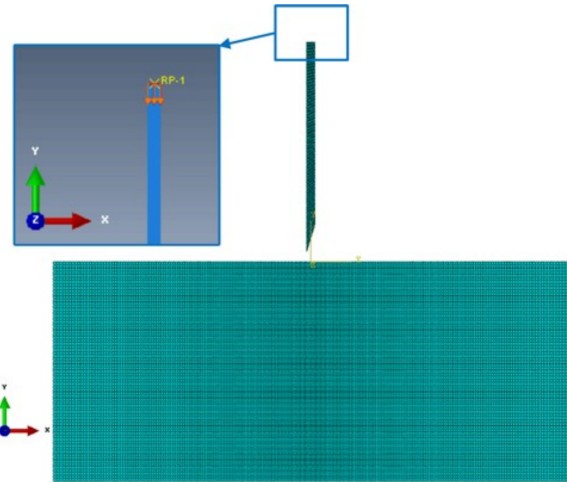

**Figure 6.** The model of tissue and a needle.

The shape and size of the needle model was established in accordance with a real situation. The mechanical property parameters of the ordinary puncture needle were obtained from the theoretical analysis value in Reference [31]. In order to observe the deformation of the needle, it was better to constrain the needle as a flexible body, and the velocity in the $y$ direction was constrained in the needle base. The parameters of the running speed $V$ and time $t$ of the needle in the insertion experiment (shown in Table 2) were consistent with the dataset in simulation. Surface-to-surface contact was defined between the needle and tissue using a kinematic contact algorithm. The friction coefficient of contact between the coating needle and the tissue during the insertion process was 0.03 and between the conventional needle and tissue was 0.42 [32].

**Table 2.** The parameters of the needle during the simulation and experiments.

| Young's Modulus (Pa) | Poisson's Ratio | Density (kg/m$^3$) | $V$ (m/s) | $t$ (s) |
|---|---|---|---|---|
| $194,020 \times 10^6$ | 0.3 | 14,800 | $8.4 \times 10^{-3}$ | 7.2 |

### 3.2. Experimental Set-Up Development

The large deformation of biological soft tissue in the puncture process made the fixation of experimental samples a problem that must be solved. Therefore, a special insertion experimental platform was designed to ensure the success of the insertion experiment in vitro. Figure 7 shows the components of the insertion experimental platform. The velocity of the needle is controlled by the displacement platform and the controller or the software connected to the displacement platform through the computer. Six-axis force and a torque sensor were mainly used to measure the force of the puncture needle during the insertion process. In order to limit the location of biological soft tissues in the insertion experiment, a tissue containment box of an adjustable size was used.

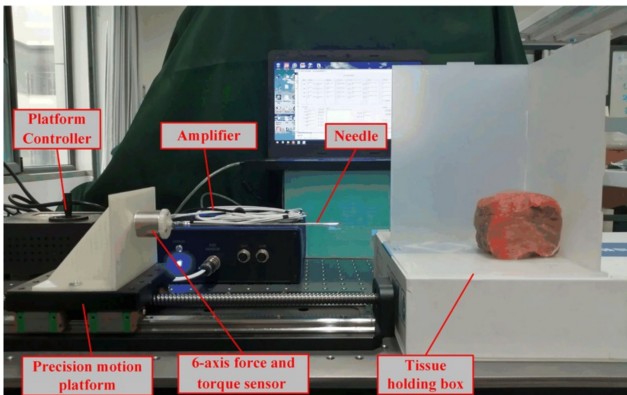

**Figure 7.** Device for verifying the experiments.

### 3.3. Model Verification

A series of needle insertion experiments were carried out on porcine loin tissue using the experiment set-up shown in Figure 7. Ten insertion experiments using the same biological soft tissue (cut into many samples) were performed. Each experimental specimen was cut into vertical blocks of 80 mm × 80 mm × 80 mm, and all surfaces were made as smooth as possible. At the beginning of needle insertion, there is a period of time when the needle pierces the dorsal membrane of the tissue. During this time, the push of the needle causes deformation of the biological soft tissue. When the deformation of the biological soft tissue unit reaches a set fracture strain value, the needle breaks the soft tissue's dorsal membrane and enters the interior of the tissue.

The insertion speed and insertion time of the needle in each experiment are controlled by the platform software. The force during each test (with the same speed and time) was recorded by a six-axis force and a torque sensor in real time. The experimental data used to verify the simulation model were obtained from the mean force of 10 insertion tests. During the insertion experiments, the pulsation of the needle may cause some variation in the $x$ and $y$ axial force measured each time. The standard deviation of the $x$ and $y$ axial force value obtained from the 10 insertion experiments is shown in Figure 8. From Figure 8, it can be seen that the disparity between each set of experimental data and the average value was not large. The dispersion degree of the $x$ axial force measured at each time point during the 10 insertion tests was approximately the same. However, the $y$ axial force was relatively discrete at the beginning of needle insertion. The experimental data had certain fluctuations, but reflected the real stress state.

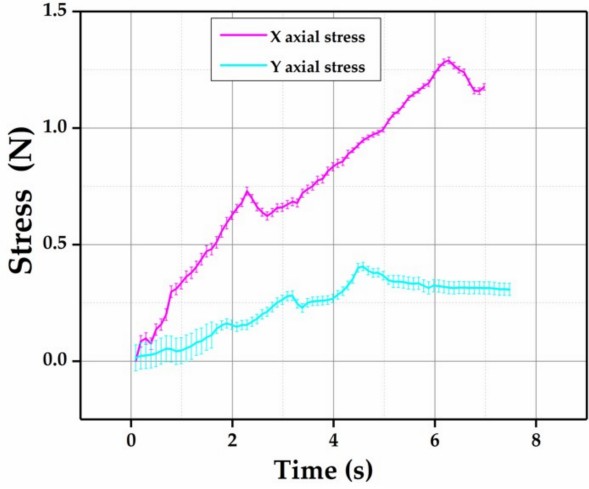

**Figure 8.** Standard deviation of data obtained from the experiment.

The insertion experimental results were used to verify the force prediction of the insertion simulation. In Figure 9a,b, the *x* and *y* axial forces measured in the insertion experiments were compared to the axial forces obtained by the insertion simulation results. The *x* axial force reached a local peak at 2.5 and 6.5 s, and at the later stage of needle insertion, the *x* axial force had a declining trend. This indicates that when the needle was inside the tissue, when the puncture depth increased, the axial reaction force of the needle decreased. The *y* axial force reached a local peak at the positions of 1.8, 3.2, and 4.7 s. Unlike the *x* axis force, the *y* axial force gradually flattened at the later stage of the puncture. This was because the *y* axial force was evenly distributed on the surface of the needle body. At a late stage, the puncture area of the needle body entering biological soft tissue increased, and even if the force increased, the measured force tended to remain unchanged.

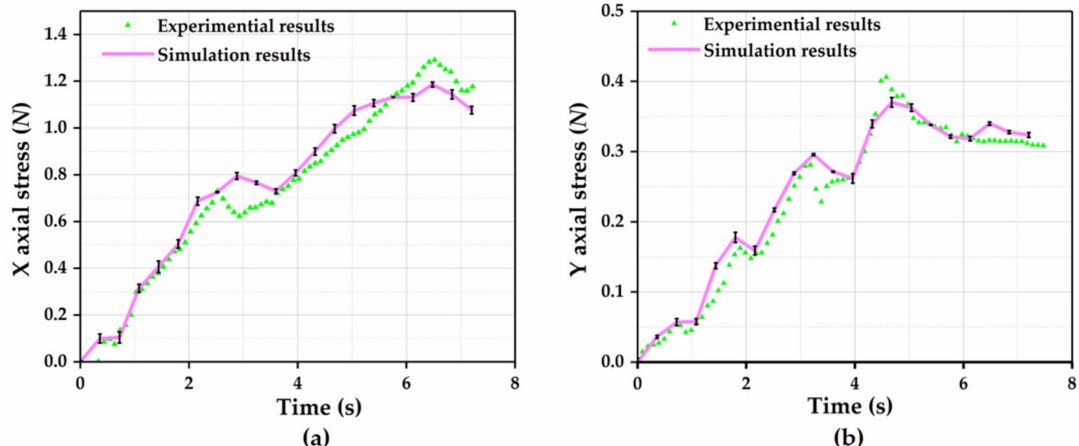

**Figure 9.** Comparison between model results and experimental results: (**a**) *x* axial force measured by a sensor in the experiment and obtained by simulation; (**b**) *y* axial force measured by a sensor in the experiment and obtained by simulation.

Since the biological soft tissue used in the experiment was uneven, two tests could not be carried out in the same place on the same sample. In order to avoid the differences caused by different puncture positions, a finite element model could be used for the analysis. The average of the *x* and *y* axial force data obtained in the model simulation was compared to the experimental data. In the standard deviation of the data, it can be seen that the data measured by simulation were of high accuracy. As can be seen from Figure 9, the experimental data were roughly consistent with the data obtained from the model simulation. The FE model could not simulate all of the actual conditions; for example, the environmental temperature and other factors were not considered. Due to simplifications in the simulation model, there was a little difference between the simulation data and experimental data. Moreover, in order to firmly connect the needle body with the six-axis force and torque sensor, there was a needle gripper in the experimental setup that may have had some influence on the complete and accurate transmission of the force. In general, however, the simulation model could show the real puncture situation. The simulation modeling method could be used to verify the differences between the puncture processes using a conventional needle and the coated needle.

## 4. Results and Discussions

To explore the effect of the coating application on the puncture process, two models of needles with a biocompatible hydrophilic superlubrication coating and a conventional needle were used. In the practical application of the biocompatible hydrophilic superlubrication coating, the high lubricity function of the coating could only be activated when the coating entered the biological soft tissue and touched the water. In the modeling process, it was assumed that the superlubrication coating would play its function initially. Thus, in the model, coatings and noncoatings were distinguished by different

friction coefficients between the needle and tissue. The results obtained from the two models will be mainly used for analysis below.

### 4.1. Strain of Biological Soft Tissue

When the maximum stresses of the needle tips were beyond the yield stress of the biological soft tissue, the tissue could be cut. A puncture biopsy is a process during which a needle continuously penetrates biological soft tissue. In a puncture biopsy, the pain received by patients has a very important relationship with the deformation of biological soft tissue. The entry of a needle at a certain speed will cause deformation of the biological soft tissue. The amount of strain on the biological soft tissue revealed the extent of damage to the tissue. In Figure 10, the strain of the biological soft tissue when fracture behavior within the biological soft tissue occurred in the model is shown. During the needle insertion, the deformation of the biological soft tissue produced in the puncture process using the coated needle was significantly lower than in the puncture process using the conventional needle. It can also be seen in Figure 10 that when the depth of the needle insertion increased, uneven burrs gradually appeared at the contact surface between the needle and the biological soft tissue. However, after applying the coating, tissue damage at the contact surface was significantly improved. Meanwhile, the application of a coating made the biological soft tissue around the needle incision smooth.

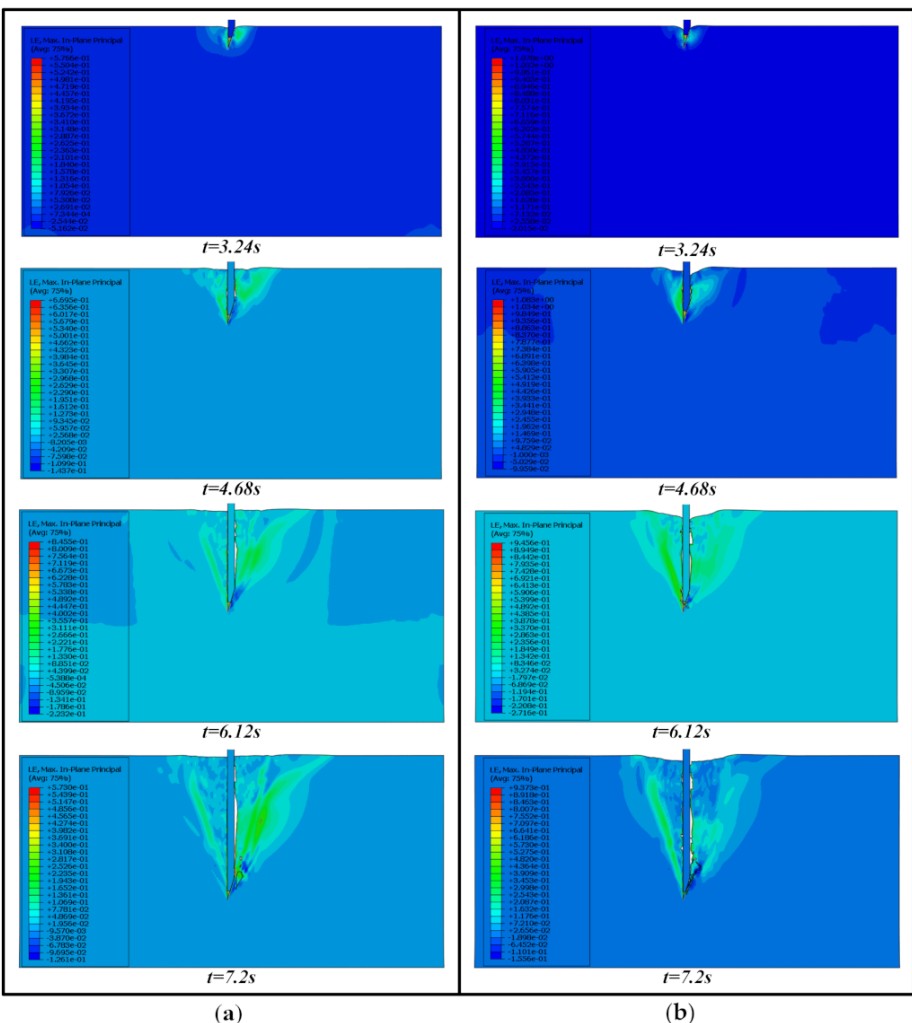

**Figure 10.** Strain contours of the biological soft tissue: (**a**) puncture process using a coated needle; (**b**) puncture process using an uncoated needle.

### 4.2. Friction and Adhesion

Tissue cutting consists of cutting, friction, and elastic forces from the surrounding tissue [33]. Friction force exists throughout the entire puncture process. The friction force of the needle in contact with the biological soft tissue is closely related to the degree of lubrication of the needle body surface. Since in the process of needle insertion, the essence of friction is the loss of energy, the lower amount of total friction-dissipated energy means less work is done to overcome the friction. Therefore, to verify the lubrication function of the biocompatible hydrophilic super-lubrication coating, the friction-dissipated energy of the entire model during needle insertion was analyzed.

The friction-dissipated energy of the entire model, which was obtained during the two simulation processes, is shown in Figure 11. The total friction-dissipated energy generated from needle insertion using the coated needle was 3.8 mJ. However, the total friction-dissipated energy of the conventional needle model was 33.9 mJ, which was more than 10 times the friction-dissipated energy generated by the coated needle model. In the case of the same needle entry path, needle speed, performance period, and biological soft tissue, less work was done to overcome the friction, meaning that there was less friction between the contact surfaces of the needle and tissue.

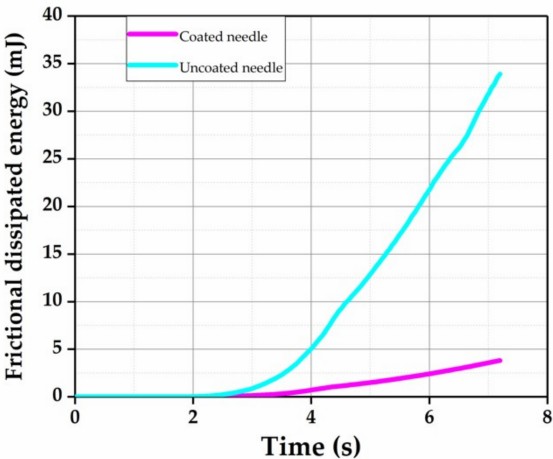

**Figure 11.** Frictional dissipated energy for the whole model.

A coated needle and conventional needle were used to carry out an in vitro needle insertion experiment. Interestingly, by observing the surfaces of the coated needle and the conventional needle after the needles exited the tissue sample, the residual biological soft tissue debris on the coated needle surface was significantly less than that on the conventional needle surface. In Figure 12, the biological soft tissue debris on the needle body surface in the two cases can be observed clearly under a biological microscope, and it is consistent with the tissue strain diagram in Figure 10. This indicates that lubrication of the coating can weaken the adhesion of biological soft tissue, which is beneficial for reducing tissue damage.

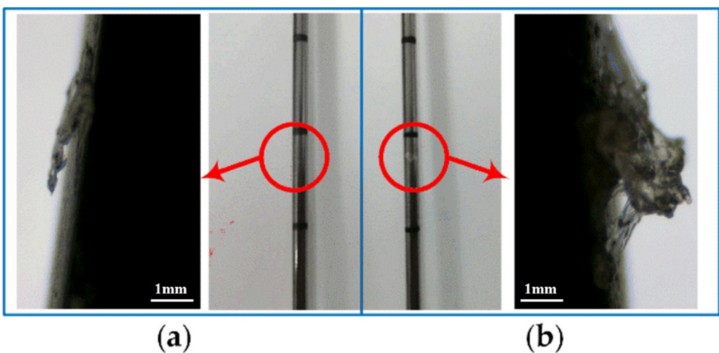

**Figure 12.** Microscopic image of the needle body surface: (**a**) coated needle; (**b**) uncoated needle.

### 4.3. Needle Deflection and Biological Soft Tissue Stress

It has been observed that larger biopsy samples can be obtained by lowering the cutting forces [34]. When a needle is injected, the needle body will deform under a reaction force. A small deformation in a needle will cause the needle to deviate from its predetermined path, which is very unfavorable in terms of accurately obtaining the pathological tissue of the examination. Hence, reducing the reaction force on a needle can help to reduce the deformation of the needle body and obtain larger biopsy samples. This ensures that the needle can reach the lesion successfully on a predetermined path and retrieve as many sample tissues as possible. Furthermore, the force on the biological soft tissue causes the patient to feel pain. When the biological soft tissue stress is small, the pain people feel is not obvious.

Figure 13 shows the maximum stress point of malformed biological soft tissues and needles for two different cases of puncture end points. Numerically, the maximum force on the coated needle was significantly lower than that on the uncoated needle, and so was the maximum force on the biological soft tissue. It can be seen that the order of magnitude of stress on the needle was larger than the stress on the biological soft tissue. However, the position of the maximum force point in both cases was roughly the same. During the insertion process, the maximum stress on the biological soft tissue was distributed at the cutting edge of the needle tip. As opposed to the stress distribution of the biological soft tissue, the maximum stress point of the needle was approximately in the middle part of the needle.

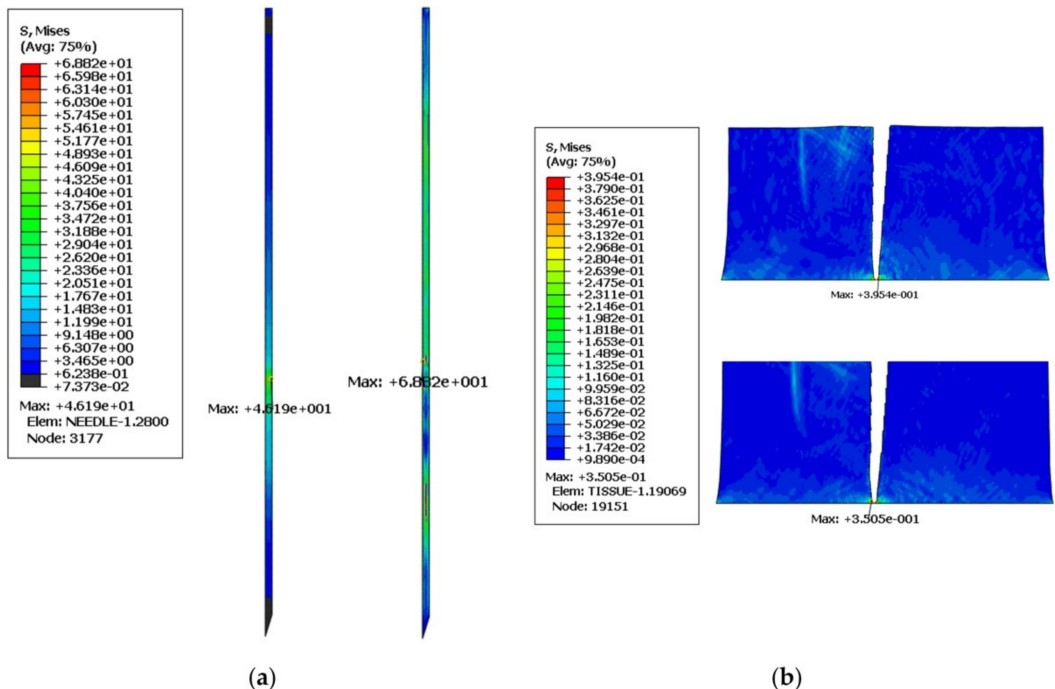

**Figure 13.** von Mises stress contours: (**a**) the maximum force on the needle at the same time point (coated needle, left, uncoated needle, right); (**b**) maximum force on the tissue at the same time point (coated needle, down, uncoated needle, up).

During the insertion process, after the needle enters the human body, the stress on the tissue causes pain. Figure 14 compares the maximum stress value of the biological soft tissue at each time point during the simulated puncture process. It can be seen in Figure 14 that the differences between the two processes in terms of the maximum force on the biological soft tissue had fluctuations. However, the process of applying a coated needle at the maximum stress point was less than that of a conventional needle. The maximum force on the biological soft tissue was significantly reduced during the puncture process with the coated needle.

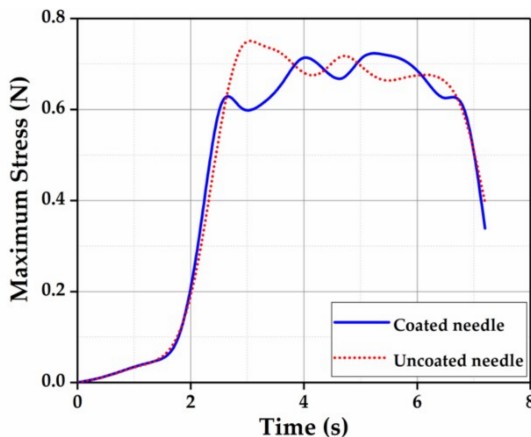

**Figure 14.** Maximum stress on the biological soft tissue.

In addition, the maximum stress point of a needle always appears in the middle of the needle body. This can easily cause the needle body to deform and change the path of a needle so that it cannot reach the lesion successfully. In Figure 15, the deformation of the points in the maximum stress concentration area of the two needles during the entire process was selected for comparison. With the deepening of the needle, the deformation of the needle increased gradually. The application of the coated needles reduced the needle deformation by approximately 7%–10%, which was beneficial for the needle insertion. In an actual puncture biopsy operation, because of accidents with manual needle insertion, minimizing the deformation of a needle is of vital importance in improving the deflection of the needle.

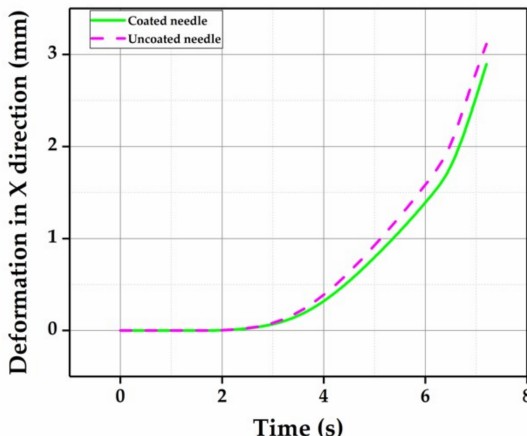

**Figure 15.** The deformation in the *x* direction at the maximum stress point.

## 5. Conclusions

The optimization of a needle body surface is of great significance in improving the process of needle insertion. In common minimally invasive puncture biopsy surgeries, the pain caused by insertion and, after the withdrawal of the needle, tissue adhered to the needle surface are the main problems that need to be solved. This study investigated the effect of a coating applied on a needle for the puncture process. A two-dimensional finite element model was developed and an experimental platform was established to verify the accuracy of the model. By using models to analyze changes in the stresses on the needle and the biological soft tissue, the total friction loss energy and strain energy of the puncture process model could be obtained. The following conclusions were reached in the study:

- The smooth incision was beneficial to the recovery of a wound caused by surgery. Extending this to the macroscopic level, the strain of the biological soft tissue corresponded with the deformation

of punctured biological soft tissue during surgery. Using a coating on a puncture needle can obviously reduce the deformation of biological soft tissue during a puncture biopsy;

- Less friction-dissipated energy means less work is done to resist friction. The coating had a significant effect on reducing the friction dissipation energy, that is, the coating had a good lubricating effect on the puncture biopsy. Reducing the friction between the needle body and the biological soft tissue can reduce patient pain and the adhesion of tissue to the needle body;

- It was obvious that the application of a coated needle had a positive effect on reducing the stress on the biological soft tissue and needle. Moreover, the effect on the reduction of stress on the needle body was obvious. Moreover, applying a coating to the surface of a needle can reduce the deformation of the needle. This is of great significance in relieving the pain of patients and inhibiting the deflection of the needle during puncture biopsy surgery.

The application of a coating is a method to optimize the surface of a needle body. The coating used in this study was a medical biocompatible hydrophilic coating that can be applied to a wide range of devices used in minimally invasive surgery to produce similar effects. In future studies, researchers can continue to explore whether other methods of surface optimization have the same effect on puncture processes. Further, different surface optimization methods can be combined to achieve multiple requirements within the same timeframe. In addition, minimally invasive surgery is becoming increasingly mechanized. Whether the optimization of a needle body surface plays a positive role in maintaining the planned puncture path of a puncture robot is worth further discussion.

**Author Contributions:** Conceptualization, Q.S.; methodology, F.G.; software, F.G.; validation, F.G.; formal analysis, Y.J.; investigation, F.G., Y.J.; resources, Q.S., Z.L., and X.H.; data curation, F.G.; writing—original draft preparation, F.G.; writing—review and editing, Q.S. and Y.J.; visualization, F.G.; supervision, Q.S. and Z.L.; project administration, Z.L.; funding acquisition, Q.S. and Z.L. All authors have read and agreed to the published version of the manuscript.

**Funding:** The authors are grateful to the Natural Science Outstanding Youth Fund of Shandong Province (Grant No. ZR2019JQ19), the Fundamental Research Funds of Shandong University (no. 2017JC027), and the Tai Shan Scholar Foundation (no. TS20130922). The authors declare that there are no conflicts of interest.

**Conflicts of Interest:** The authors declare no conflicts of interest.

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
