# Peer review of "Influence of a Biocompatible Hydrophilic Needle Surface Coating on a Puncture Biopsy Process for Biomedical Applications"

_coatings, doi:10.3390/coatings10020178_

Round 1
Reviewer 1 Report
The manuscript titled "Influence of biocompatible hydrophilic needle surface coating on puncture biopsy process for biomedical applications" is a work of high significance which can result in the development of novel coatings for needles that may be able to ease the pain of patients. Therefore, such developments are noble and require strong attention. This valuable work further progresses the field of coatings and medicine. Findings are of high interest and importance; results are well explained and interpretations are good. The major drawback of the paper is a language issue. I tried my best to suggest improvements throughout the text. Also, there are some comments regarding the confidence in the results, as this is of extreme significance in medical applications. There are also some unclarities in the text and some language issues that should be addressed. Once these are corrected, I strongly recommend that this paper is accepted for publication. See my comments:
Are these coating materials not harmful in contact with bodily fluids? Not carcinogenic? Check safety data sheets
- Line 16: …is a widely used…
- Line 17: …body is always in direct contact with a biological soft tissue…
- Line 18: …damage may occur…
- Line 18: Not clear what is meant with “time to time”. Rephrase.
- Line 19: …surface, especially, lubrication of a needle…
- Line 20: …applied onto…
- Line 23: …accuracy of the model…
- Line 23: …of a coated needle and a conventional…
- Line 24: needle, the influence…
- Line 24: …process was obtained.
- Line 32: Not clear what is meant with “local treatment play”. Rephrase. What plays? Puncture needle?
- Line 34: being less traumatic?
- Line 37: …diagnosis of a pathological…
- Line 39: …have a detailed…
- Line 40: … about the puncture…
- Line 42: analyse the soft tissue deformation during the insertion process. They simulated …
- Line 43: …point in a biological soft tissue …
- Line 46: …body and the…
- Line 50: …tip of the needle…
- Line 51: …on the path because the nodes on the path…
- Line 52: Not clear what is meant with “used gel replace the soft biological tissue”. Rephrase.
- Line 52: Expand FE – Finite Element. First time in the text.
- Line 53: simulate a needle-tissue interaction during the needle insertion …
- Line 53: …used a linear elastic model…
- Line 54: Why cannot represent the real situation? Explain why a bit more.
- Line 63: …into a brain tissue…
- Line 64: Expand DBS.
- Line 69: Expand PLGA
- Line 72: How does it allow to record signals for a long time if it degrades quickly? Not clear. Explain.
- Line 75: …tissue to the needle body …
- Line 97: “passed the national evaporation test”. Provide a reference – refer to a source. Also, you should compare to an international standard, i.e. ISO or ASTM, if there is one. Does your case also pass the international standard requirements and test?
- Line 100: …micron-scale, it is difficult…
- Line 105: “and the reflection is serious”. Rephrase
- Figure 2: …of the needle surface…
- Figure 2: …of the coated needle…
- Figure 2: …of the uncoated needle…
- Line 111: Check “celiac”
- Line 112: …experimental samples, muscle…
- Line 113: Check “adventitia”
- Line 118: …Meanwhile it is the same as the material used in the uniaxial tension test.
- Line 135: …materials [Reference?].
- Line 136: …behavior of a biological …
- Line 137: function [Reference?]. …
- Line 155: Are these physical properties or fitting parameters? Explain more.
- Line 164: fitted onto a straight line. …
- Line 164: “C10 is the intercept of this line and C01 is the slope of this line”. Explain here a bit more what does it mean from a physical point of view. What is a physical interpretation of those parameters?
- Table 1: Check units. Is it typical to use ton/mm3? Why not use SI system?
- Table 1: Give units for C10 and C01.
- Table 1: 1.12E-009 should be written as 1.12∙10-9?
- Line 168: Expand COF – coefficient of friction. First time in the text.
- Line 170: …properties of the needle. …
- Line 171: …the needle and the tissue …
- Line 172: …contact between the coated needle and the tissue… OR …coating of the needle…
- Line 173: …between the coated needle
- Line 174: and the biological soft tissue…
- Line 177: A flat piece of a biological soft …
- Line 177: … pressure between the
- Line 182: After the analysis, it was found…
- Line 183: … Therefore, the friction coefficient…
- Line 184: …are fitted onto a straight…
- Line 185: Not clear what is meant. Rephrase
- Line 186: …after the insertion…
- Line 187: because of the complex mechanical conditions of a biological soft tissue, including …
- Line 188: How much error is acceptable? Provide statistical errors and a reference to a standard practice if possible.
- Line 189: …provider [Reference?].
- Figure 5: Explain in the text (just a sentence or a few) what does it signify that the measured Fx = f(N) is well linear.
- Line 196: …Table 1 …
- Figure 6: … and needle. (Period forgotten at the end of the title).
- Line 208: …analysis value from [31].
- Line 209: …deformation of the needle…
- Line 209: …as a flexible body,
- Line 210: What is an edge of the needle? Rephrase
- Line 211: speed V and time t of the needle…
- Line 213: …contact between the coated needle and the tissue during the insertion…
- Line 214: …and between a conventional needle and the tissue…
- Line 214: Provide statistical errors, standard deviations for 0.03 and 0.42.
- Table 2: ton or tonne? It is different. Check units here too! Not SI system?
- Table 2: Where are these parameters taken from? Give reference
- Line 216: Experimental setup development
- Line 219: “in vitro” should be in italic.
- Line 231: … beginning of the needle
- Line 232: …punctures the dorsal membrane of the tissue. ??? Check
- Line 233: needle pierces…
- Line 233: …push of the needle caused deformation of the biological soft…
- Line 235: …the soft tissue’s dorsal membrane and enters the interior of the tissue.
- Line 236: …time of the needle…
- Line 237: … by a 6-axis…
- Line 239: … During the insertion experiments…
- Line 240: cause some variation in …
- Figure 8: Very good! Also answers part of my comment regarding statistical errors. I suggest to make another graph 8(b), which is a manifestation of this one 8(a), but would show COF over time and its standard deviations. Add one graph beside the other.
- Line 250: “regional”? Perhaps “local” is meant
- Line 252: “inside the organization”. Not clear what is meant. Rephrase.
- Line 253: “regional”? Perhaps “local” is meant
- Line 261: … obtained from the model simulation. …
- Line 262: How crucial is that environment temperature was not considered? Can you comment a bit more on the influence of temperature? Does this affect COF of the coating?
- Line 263: …in the simulation model…
- Line 268: …using a conventional needle and the coated needle.
- Figure 9: Show error bars (Standard deviations) of the experimental data curves. Perhaps, this may give even more confidence in your model
- Line 277: What would be COF without water? Can this be potentially a problem, for instance, for dry skin?
- Line 284: …process during which needle is continuously cutting (penetrating?) a biological soft tissue. …
- Line 301: … force of the needle …
- Line 302: in contact with the biological …
- Line 302: … lubrication of the needle body …
- Line 303: … needle insertion, the essence …
- Line 307: … two simulation processes …
- Line 308: … using the coated needle …
- Line 309: … of a conventional …
- Line 309: Provide standard deviations for 3.8 mJ and 33.9 mJ.
- Line 316: “in vitro” should be in italic
- Line 317: … surfaces of the coated needle and a conventional …
- Line 319: … that on a conventional …
- Line 320: “Obviously”. Perhaps better use clearly?
- Line 321: …that lubrication of the …
- Lines 322-323: “Which will beneficial for improving tissue damage”. Not clear. Rephrase.
- Line 329: … cause needle to deviate from …
- Lines 332-333: “as many as possible”. Not clear. Rephrase.
- Line 334: Not clear. Rephrase.
- Line 337: … force on the biological …
- Line 338: … can be seen that the …
- Line 342: … middle part of the needle.
- Figure 13: …force on the needle at the same point in time …
- Figure 13: …on the tissue at the same point in time …
- Line 347: “intuitive pain to patients”. Check this sentence. Does it make sense? Perhaps explain. Any reference?
- Line 351: … that of a conventional…
- Line 351: … force on the biological …
- Line 352: … with the coated needle.
- Line 355: …point of a needle…
- Line 355: …in the middle of the needle body …
- Line 356: …path of a needle …
- Line 359: …of a needle increases…
- Line 360: …for the needle insertion.
- Line 362: deformation of a needle…
- Line 362: …deflection of the needle.
- Figure 15: …at the maximum stress point.
- Line 366: …of a needle body …
- Line 368: insertion and, after withdrawing the needle, tissue adhered to the needle surface are the problems to be solved.
- Line 369: This study investigated the effect of a coating (applied on the needle) on the puncture process.
- Line 371: … of the model.
- Line 371: …stresses on the needle
- Line 372: and the biological soft tissue…
- Line 372: …of the puncture process model.
- Line 373: The following conclusions were reached in the study:
- Line 374: …recovery of a wound…
- Line 376: … coating on a puncture needle…
- Line 387: …surface of a needle body.
- Line 393: …optimization of a needle body surface…
// Strong accept after the comments have been addressed.
Reviewer 2 Report
The manuscript entitled “Influence of biocompatible hydrophilic needle surface coating on puncture biopsy process for biomedical applications” discussed hydrophilic needle surface coating for human tissue biopsy application or other biomedical application. This manuscript can be improved after major revision. The issues are listed as below.
- Author must include data on hydrophilicity experiment on this kind of coating on material using well know experiments such as contact angle etc…
- Author must include theoretically and computational modeling to prove functionalization or hydorphilicity on this kind of materials.
- Author must include data on damage on human or animal tissue using various methods such as histopathology, tissue staining or cell viability etc.
Round 2
Reviewer 2 Report
I recommend to accept this manuscript